# Deep Hyperspherical Learning

**Weiyang Liu[1], Yan-Ming Zhang[2], Xingguo Li[3,1], Zhiding Yu[4], Bo Dai[1], Tuo Zhao[1], Le Song[1]**
[1]Georgia Institute of Technology   [2]Institute of Automation, Chinese Academy of Sciences
[3]University of Minnesota   [4]Carnegie Mellon University
{wyliu,tourzhao}@gatech.edu, ymzhang@nlpr.ia.ac.cn, lsong@cc.gatech.edu

## Abstract

Convolution as inner product has been the founding basis of convolutional neural networks (CNNs) and the key to end-to-end visual representation learning. Benefiting from deeper architectures, recent CNNs have demonstrated increasingly strong representation abilities. Despite such improvement, the increased depth and larger parameter space have also led to challenges in properly training a network. In light of such challenges, we propose hyperspherical convolution (SphereConv), a novel learning framework that gives angular representations on hyperspheres. We introduce SphereNet, deep hyperspherical convolution networks that are distinct from conventional inner product based convolutional networks. In particular, SphereNet adopts SphereConv as its basic convolution operator and is supervised by generalized angular softmax loss - a natural loss formulation under SphereConv. We show that SphereNet can effectively encode discriminative representation and alleviate training difficulty, leading to easier optimization, faster convergence and comparable (even better) classification accuracy over convolutional counterparts. We also provide some theoretical insights for the advantages of learning on hyperspheres. In addition, we introduce the learnable SphereConv, i.e., a natural improvement over prefixed SphereConv, and SphereNorm, i.e., hyperspherical learning as a normalization method. Experiments have verified our conclusions.

## 1 Introduction

Recently, deep convolutional neural networks have led to significant breakthroughs on many vision problems such as image classification [9, 18, 19, 6], segmentation [3, 13, 1], object detection [3, 16], etc. While showing stronger representation power over many conventional hand-crafted features, CNNs often require a large amount of training data and face certain training difficulties such as overfitting, vanishing/exploding gradient, covariate shift, etc. The increasing depth of recently proposed CNN architectures have further aggravated the problems.

To address the challenges, regularization techniques such as dropout [9] and orthogonality parameter constraints [21] have been proposed. Batch normalization [8] can also be viewed as an implicit regularization to the network, by normalizing each layer's output distribution. Recently, deep residual learning [6] emerged as a promising way to overcome vanishing gradients in deep networks. However, [20] pointed out that residual networks (ResNets) are essentially an exponential ensembles of shallow networks where they avoid the vanishing/exploding gradient problem but do not provide direct solutions. As a result, training an ultra-deep network still remains an open problem. Besides vanishing/exploding gradient, network optimization is also very sensitive to initialization. Finding better initializations is thus widely studied [5, 14, 4]. In general, having a large parameter space is double-edged considering the benefit of representation power and the associated training difficulties. Therefore, proposing better learning frameworks to overcome such challenges remains important.

In this paper, we introduce a novel convolutional learning framework that can effectively alleviate training difficulties, while giving better performance over dot product based convolution. Our idea

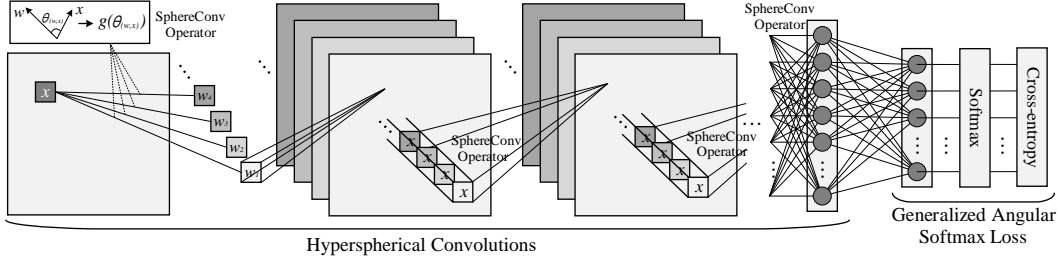

Figure 1: Deep hyperspherical convolutional network architecture.

is to project parameter learning onto unit hyperspheres, where layer activations only depend on the geodesic distance between kernels and input signals[1] instead of their inner products. To this end, we propose the SphereConv operator as the basic module for our network layers. We also propose softmax losses accordingly under such representation framework. Specifically, the proposed softmax losses supervise network learning by also taking the SphereConv activations from the last layer instead of inner products. Note that the geodesic distances on a unit hypersphere is the angles between inputs and kernels. Therefore, the learning objective is essentially a function of the input angles and we call it generalized angular softmax loss in this paper. The resulting architecture is the hyperspherical convolutional network (SphereNet), which is shown in Fig. 1.

Our key motivation to propose SphereNet is that angular information matters in convolutional representation learning. We argue this motivation from several aspects: training stability, training efficiency, and generalization power. SphereNet can also be viewed as an implicit regularization to the network by normalizing the activation distributions. The weight norm is no longer important since the entire network operates only on angles. And as a result, the $\ell_2$ weight decay is also no longer needed in SphereNet. SphereConv to some extent also alleviates the covariate shift problem [8]. The output of SphereConv operators are bounded from $-1$ to $1$ (0 to 1 if considering ReLU), which makes the variance of each output also bounded.

Our second intuition is that angles preserve the most abundant discriminative information in convolutional learning. We gain such intuition from 2D Fourier transform, where an image is decomposed by the combination of a set of templates with magnitude and phase information in 2D frequency domain. If one reconstructs an image with original magnitudes and random phases, the resulting images are generally not recognizable. However, if one reconstructs the image with random magnitudes and original phases. The resulting images are still recognizable. It shows that the most important structural information in an image for visual recognition is encoded by phases. This fact inspires us to project the network learning into angular space. In terms of low-level information, SphereConv is able to preserve the shape, edge, texture and relative color. SphereConv can learn to selectively drop the color depth but preserve the RGB ratio. Thus the semantic information of an image is preserved.

SphereNet can also be viewed as a non-trivial generalization of [12, 11]. By proposing a loss that discriminatively supervises the network on a hypersphere, [11] achieves state-of-the-art performance on face recognition. However, the rest of the network remains a conventional convolution network. In contrast, SphereNet not only generalizes the hyperspherical constraint to every layer, but also to different nonlinearity functions of input angles. Specifically, we propose three instances of SphereConv operators: linear, cosine and sigmoid. The sigmoid SphereConv is the most flexible one with a parameter controlling the shape of the angular function. As a simple extension to the sigmoid SphereConv, we also present a learnable SphereConv operator. Moreover, the proposed generalized angular softmax (GA-Softmax) loss naturaly generalizes the angular supervision in [11] using the SphereConv operators. Additionally, the SphereConv can serve as a normalization method that is comparable to batch normalization, leading to an extension to spherical normalization (SphereNorm).

SphereNet can be easily applied to other network architectures such as GoogLeNet [19], VGG [18] and ResNet [6]. One simply needs to replace the convolutional operators and the loss functions with the proposed SphereConv operators and hyperspherical loss functions. In summary, SphereConv can be viewed as an alternative to the original convolution operators, and serves as a new measure of correlation. SphereNet may open up an interesting direction to explore the neural networks. We ask the question *whether inner product based convolution operator is an optimal correlation measure for all tasks?* Our answer to this question is likely to be "no".

# 2 Hyperspherical Convolutional Operator

## 2.1 Definition

The convolutional operator in CNNs is simply a linear matrix multiplication, written as $\mathcal{F}(\boldsymbol{w}, \boldsymbol{x}) = \boldsymbol{w}^\top \boldsymbol{x} + b_{\mathcal{F}}$ where $\boldsymbol{w}$ is a convolutional filter, $\boldsymbol{x}$ denotes a local patch from the bottom feature map and $b_{\mathcal{F}}$ is the bias. The matrix multiplication here essentially computes the similarity between the local patch and the filter. Thus the standard convolution layer can be viewed as patch-wise matrix multiplication. Different from the standard convolutional operator, the hyperspherical convolutional (SphereConv) operator computes the similarity on a hypersphere and is defined as:

$$\mathcal{F}_s(\boldsymbol{w}, \boldsymbol{x}) = g(\theta_{(\boldsymbol{w}, \boldsymbol{x})}) + b_{\mathcal{F}_s}, \tag{1}$$

where $\theta_{(\boldsymbol{w}, \boldsymbol{x})}$ is the angle between the kernel parameter $\boldsymbol{w}$ and the local patch $\boldsymbol{x}$. $g(\theta_{(\boldsymbol{w}, \boldsymbol{x})})$ indicates a function of $\theta_{(\boldsymbol{w}, \boldsymbol{x})}$ (usually a monotonically decreasing function), and $b_{\mathcal{F}_s}$ is the bias. To simplify analysis and discussion, the bias terms are usually left out. The angle $\theta_{(\boldsymbol{w}, \boldsymbol{x})}$ can be interpreted as the geodesic distance (arc length) between $\boldsymbol{w}$ and $\boldsymbol{x}$ on a unit hypersphere. In contrast to the convolutional operator that works in the entire space, SphereConv only focuses on the angles between local patches and the filters, and therefore operates on the hypersphere space. In this paper, we present three specific instances of the SphereConv Operator. To facilitate the computation, we constrain the output of SphereConv operators to $[-1, 1]$ (although it is not a necessary requirement).

**Linear SphereConv**. In linear SphereConv operator, $g$ is a linear function of $\theta_{(\boldsymbol{w}, \boldsymbol{x})}$, with the form:

$$g(\theta_{(\boldsymbol{w}, \boldsymbol{x})}) = a\theta_{(\boldsymbol{w}, \boldsymbol{x})} + b, \tag{2}$$

where $a$ and $b$ are parameters for the linear SphereConv operator. In order to constrain the output range to $[0, 1]$ while $\theta_{(\boldsymbol{w}, \boldsymbol{x})} \in [0, \pi]$, we use $a = -\frac{2}{\pi}$ and $b = 1$ (not necessarily optimal design).

**Cosine SphereConv**. The cosine SphereConv operator is a non-linear function of $\theta_{(\boldsymbol{w}, \boldsymbol{x})}$, with its $g$ being the form of

$$g(\theta_{(\boldsymbol{w}, \boldsymbol{x})}) = \cos(\theta_{(\boldsymbol{w}, \boldsymbol{x})}), \tag{3}$$

which can be reformulated as $\frac{\boldsymbol{w}^T \boldsymbol{x}}{\|\boldsymbol{w}\|_2 \|\boldsymbol{x}\|_2}$. Therefore, it can be viewed as a doubly normalized convolutional operator, which bridges the SphereConv operator and convolutional operator.

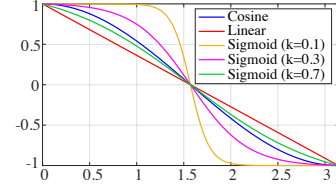

Figure 2: SphereConv operators.

**Sigmoid SphereConv**. The Sigmoid SphereConv operator is derived from the Sigmoid function and its $g$ can be written as

$$g(\theta_{(\boldsymbol{w}, \boldsymbol{x})}) = \frac{1 + \exp(-\frac{\pi}{2k})}{1 - \exp(-\frac{\pi}{2k})} \cdot \frac{1 - \exp\left(\frac{\theta_{(\boldsymbol{w}, \boldsymbol{x})}}{k} - \frac{\pi}{2k}\right)}{1 + \exp\left(\frac{\theta_{(\boldsymbol{w}, \boldsymbol{x})}}{k} - \frac{\pi}{2k}\right)}, \tag{4}$$

where $k > 0$ is the parameter that controls the curvature of the function. While $k$ is close to 0, $g(\theta_{(\boldsymbol{w}, \boldsymbol{x})})$ will approximate the step function. While $k$ becomes larger, $g(\theta_{(\boldsymbol{w}, \boldsymbol{x})})$ is more like a linear function, i.e., the linear SphereConv operator. Sigmoid SphereConv is one instance of the parametric SphereConv family. With more parameters being introduced, the parametric SphereConv can have richer representation power. To increase the flexibility of the parametric SphereConv, we will discuss the case where these parameters can be jointly learned via back-prop later in the paper.

## 2.2 Optimization

The optimization of the SphereConv operators is nearly the same as the convolutional operator and also follows the standard back-propagation. Using the chain rule, we have the gradient of the SphereConv with respect to the weights and the feature input:

$$\frac{\partial g(\theta_{(\boldsymbol{w}, \boldsymbol{x})})}{\partial \boldsymbol{w}} = \frac{\partial g(\theta_{(\boldsymbol{w}, \boldsymbol{x})})}{\partial \theta_{(\boldsymbol{w}, \boldsymbol{x})}} \cdot \frac{\partial \theta_{(\boldsymbol{w}, \boldsymbol{x})}}{\partial \boldsymbol{w}}, \quad \frac{\partial g(\theta_{(\boldsymbol{w}, \boldsymbol{x})})}{\partial \boldsymbol{x}} = \frac{\partial g(\theta_{(\boldsymbol{w}, \boldsymbol{x})})}{\partial \theta_{(\boldsymbol{w}, \boldsymbol{x})}} \cdot \frac{\partial \theta_{(\boldsymbol{w}, \boldsymbol{x})}}{\partial \boldsymbol{x}}. \tag{5}$$

For different SphereConv operators, both $\frac{\partial \theta_{(\boldsymbol{w}, \boldsymbol{x})}}{\partial \boldsymbol{w}}$ and $\frac{\partial \theta_{(\boldsymbol{w}, \boldsymbol{x})}}{\partial \boldsymbol{x}}$ are the same, so the only difference lies in the $\frac{\partial g(\theta_{(\boldsymbol{w}, \boldsymbol{x})})}{\partial \theta_{(\boldsymbol{w}, \boldsymbol{x})}}$ part. For $\frac{\partial \theta_{(\boldsymbol{w}, \boldsymbol{x})}}{\partial \boldsymbol{w}}$, we have

$$\frac{\partial \theta_{(\boldsymbol{w}, \boldsymbol{x})}}{\partial \boldsymbol{w}} = \frac{\partial \arccos\left(\frac{\boldsymbol{w}^T \boldsymbol{x}}{\|\boldsymbol{w}\|_2 \|\boldsymbol{x}\|_2}\right)}{\partial \boldsymbol{w}}, \quad \frac{\partial \theta_{(\boldsymbol{w}, \boldsymbol{x})}}{\partial \boldsymbol{x}} = \frac{\partial \arccos\left(\frac{\boldsymbol{w}^T \boldsymbol{x}}{\|\boldsymbol{w}\|_2 \|\boldsymbol{x}\|_2}\right)}{\partial \boldsymbol{x}}, \tag{6}$$

which are straightforward to compute and therefore neglected here. Because $\frac{\partial g(\theta_{(\boldsymbol{w}, \boldsymbol{x})})}{\partial \theta_{(\boldsymbol{w}, \boldsymbol{x})}}$ for the linear SphereConv, the cosine SphereConv and the Sigmoid SphereConv are $a$, $-\sin(\theta_{(\boldsymbol{w}, \boldsymbol{x})})$ and $\frac{-2 \exp(\theta_{(\boldsymbol{w}, \boldsymbol{x})}/k - \pi/2k)}{k(1 + \exp(\theta_{(\boldsymbol{w}, \boldsymbol{x})}/k - \pi/2k))^2}$ respectively, all these partial gradients can be easily computed.

## 2.3 Theoretical Insights

We provide a fundamental analysis for the cosine SphereConv operator in the case of linear neural network to justify that the SphereConv operator can improve the conditioning of the problem. In specific, we consider one layer of linear neural network, where the observation is $\boldsymbol{F} = \boldsymbol{U}^*\boldsymbol{V}^{*\top}$ (ignore the bias), $\boldsymbol{U}^* \in \mathbb{R}^{n \times k}$ is the weight, and $\boldsymbol{V}^* \in \mathbb{R}^{m \times k}$ is the input that embeds weights from previous layers. Without loss of generality, we assume the columns satisfying $\|\boldsymbol{U}_{i,:}\|_2 = \|\boldsymbol{V}_{j,:}\|_2 = 1$ for all $i = 1, \dots, n$ and $j = 1, \dots, m$, and consider

$$\min_{\boldsymbol{U} \in \mathbb{R}^{n \times k}, \boldsymbol{V} \in \mathbb{R}^{m \times k}} \mathcal{G}(\boldsymbol{U}, \boldsymbol{V}) = \tfrac{1}{2}\|\boldsymbol{F} - \boldsymbol{U}\boldsymbol{V}^\top\|_{\mathrm{F}}^2. \tag{7}$$

This is closely related with the matrix factorization and (7) can be also viewed as the expected version for the matrix sensing problem [10]. The following lemma demonstrates a critical scaling issue of (7) for $\boldsymbol{U}$ and $\boldsymbol{V}$ that significantly deteriorate the conditioning without changing the objective of (7).

**Lemma 1.** Consider a pair of global optimal points $\boldsymbol{U}, \boldsymbol{V}$ satisfying $\boldsymbol{F} = \boldsymbol{U}\boldsymbol{V}^\top$ and $\mathrm{Tr}(\boldsymbol{V}^\top\boldsymbol{V} \otimes \boldsymbol{I}_n) \leq \mathrm{Tr}(\boldsymbol{U}^\top\boldsymbol{U} \otimes \boldsymbol{I}_m)$. For any real $c > 1$, let $\widetilde{\boldsymbol{U}} = c\boldsymbol{U}$ and $\widetilde{\boldsymbol{V}} = \boldsymbol{V}/c$, then we have $\kappa(\nabla^2\mathcal{G}(\widetilde{\boldsymbol{U}}, \widetilde{\boldsymbol{V}})) = \Omega(c^2\kappa(\nabla^2\mathcal{G}(\boldsymbol{U}, \boldsymbol{V})))$, where $\kappa = \frac{\lambda_{\max}}{\lambda_{\min}}$ is the restricted condition number with $\lambda_{\max}$ being the largest eigenvalue and $\lambda_{\min}$ being the smallest nonzero eigenvalue.

Lemma 1 implies that the conditioning of the problem (7) at a unbalanced global optimum scaled by a constant $c$ is $\Omega(c^2)$ times larger than the conditioning of the problem at a balanced global optimum. Note that $\lambda_{\min} = 0$ may happen, thus we consider the restricted condition here. Similar results hold beyond global optima. This is an undesired geometric structure, which further leads to slow and unstable optimization procedures, e.g., using stochastic gradient descent (SGD). This motivates us to consider the SphereConv operator discussed above, which is equivalent to projecting data onto the hypersphere and leads to a better conditioned problem.

Next, we consider our proposed cosine SphereConv operator for one-layer of the linear neural network. Based on our previous discussion on SphereConv, we consider an equivalent problem:

$$\min_{\boldsymbol{U} \in \mathbb{R}^{n \times k}, \boldsymbol{V} \in \mathbb{R}^{m \times k}} \mathcal{G}_S(\boldsymbol{U}, \boldsymbol{V}) = \tfrac{1}{2}\|\boldsymbol{F} - \boldsymbol{D}_{\boldsymbol{U}}\boldsymbol{U}\boldsymbol{V}^\top\boldsymbol{D}_{\boldsymbol{V}}\|_{\mathrm{F}}^2, \tag{8}$$

where $\boldsymbol{D}_{\boldsymbol{U}} = \mathrm{diag}\big(\frac{1}{\|\boldsymbol{U}_{1,:}\|_2}, \dots, \frac{1}{\|\boldsymbol{U}_{n,:}\|_2}\big) \in \mathbb{R}^{n \times n}$ and $\boldsymbol{D}_{\boldsymbol{V}} = \mathrm{diag}\big(\frac{1}{\|\boldsymbol{V}_{1,:}\|_2}, \dots, \frac{1}{\|\boldsymbol{V}_{m,:}\|_2}\big) \in \mathbb{R}^{m \times m}$ are diagonal matrices. We provide an analogous result to Lemma 1 for (8) .

**Lemma 2.** For any real $c > 1$, let $\widetilde{\boldsymbol{U}} = c\boldsymbol{U}$ and $\widetilde{\boldsymbol{V}} = \boldsymbol{V}/c$, then we have $\lambda_i(\nabla^2\mathcal{G}_S(\widetilde{\boldsymbol{U}}, \widetilde{\boldsymbol{V}})) = \lambda_i(\nabla^2\mathcal{G}_S(\boldsymbol{U}, \boldsymbol{V}))$ for all $i \in [(n + m)k] = \{1, 2, \dots, (n + m)k\}$ and $\kappa(\nabla^2\mathcal{G}(\widetilde{\boldsymbol{U}}, \widetilde{\boldsymbol{V}})) = \kappa(\nabla^2\mathcal{G}(\boldsymbol{U}, \boldsymbol{V}))$, where $\kappa$ is defined as in Lemma 1.

We have from Lemma 2 that the issue of increasing condition caused by the scaling is eliminated by the SphereConv operator in the entire parameter space. This enhances the geometric structure over (7), which further results in improved convergence of optimization procedures. If we extend the result from one layer to multiple layers, the scaling issue propagates. Roughly speaking, when we train $N$ layers, in the worst case, the conditioning of the problem can be $c^N$ times worse with a scaling factor $c > 1$. The analysis is similar to the one layer case, but the computation of the Hessian matrix and associated eigenvalues are much more complicated. Though our analysis is elementary, we provide an important insight and a straightforward illustration of the advantage for using the SphereConv operator. The extension to more general cases, e..g, using nonlinear activation function (e.g., ReLU), requires much more sophisticated analysis to bound the eigenvalues of Hessian for objectives, which is deferred to future investigation.

## 2.4 Discussion

**Comparison to convolutional operators**. Convolutional operators compute the inner product between the kernels and the local patches, while the SphereConv operators compute a function of the angle between the kernels and local patches. If we normalize the convolutional operator in terms of both $\boldsymbol{w}$ and $\boldsymbol{x}$, then the normalized convolutional operator is equivalent to the cosine SphereConv operator. Essentially, they use different metric spaces. Interestingly, SphereConv operators can also be interpreted as a function of the Geodesic distance on a unit hypersphere.

**Extension to fully connected layers**. Because the fully connected layers can be viewed as a special convolution layer with the kernel size equal to the input feature map, the SphereConv operators could be easily generalized to the fully connected layers. It also indicates that SphereConv operators could be used not only to deep CNNs, but also to linear models like logistic regression, SVM, etc.

**Network Regularization**. Because the norm of weights is no longer crucial, we stop using the $\ell_2$ weight decay to regularize the network. SphereNets are learned on hyperspheres, so we regularize the network based on angles instead of norms. To avoid redundant kernels, we want the kernels uniformly spaced around the hypersphere, but it is difficult to formulate such constraints. As a tradeoff, we encourage the orthogonality. Given a set of kernels $\boldsymbol{W}$ where the $i$-th column $\boldsymbol{W}_i$ is the weights of the $i$-th kernel, the network will also minimize $\|\boldsymbol{W}^\top\boldsymbol{W} - \boldsymbol{I}\|_F^2$ where $\boldsymbol{I}$ is an identity matrix.

**Determining the optimal SphereConv.** In practice, we could treat different types of SphereConv as a hyperparameter and use the cross validation to determine which SphereConv is the most suitable one. For sigmoid SphereConv, we could also use the cross validation to determine its hyperparameter $k$. In general, we need to specify a SphereConv operator before using it, but prefixing a SphereConv may not be an optimal choice (even using cross validation). What if we treat the hyperparameter $k$ in sigmoid SphereConv as a learnable parameter and use the back-prop to learn it? Following this idea, we further extend sigmoid SphereConv to a learnable SphereConv in the next subsection.

**SphereConv as normalization.** Because SphereConv could partially address the covariate shift, it could also serve as a normalization method similar to batch normalization. Differently, SphereConv normalizes the network in terms of feature map and kernel weights, while batch normalization is for the mini-batches. Thus they do not contradict with each other and can be used simultaneously.

## 2.5  Extension: Learnable SphereConv and SphereNorm

**Learnable SphereConv.** It is a natrual idea to replace the current prefixed SphereConv with a learnable one. There will be plenty of parametrization choices for the SphereConv to be learnable, and we present a very simple learnable SphereConv operator based on the sigmoid SphereConv. Because the sigmoid SphereConv has a hyperparameter $k$, we could treat it as a learnable parameter that can be updated by back-prop. In back-prop, $k$ is updated using $k^{t+1} = k^t + \eta\frac{\partial L}{\partial k}$ where $t$ denotes the current iteration index and $\frac{\partial L}{\partial k}$ can be easily computed by the chain rule. Usually, we also require $k$ to be positive. The learning of $k$ is in fact similar to the parameter learning in PReLU [5].

**SphereNorm: hyperspherical learning as a normalization method.** Similar to batch normalization (BatchNorm), we note that the hyperspherical learning can also be viewed as a way of normalization, because SphereConv constrain the output value in $[-1, 1]$ ($[0, 1]$ after ReLU). Different from BatchNorm, SphereNorm normalizes the network based on spatial information and the weights, so it has nothing to do with the mini-batch statistic. Because SphereConv normalize both the input and weights, it could avoid covariate shift due to large weights and large inputs while BatchNorm could only prevent covariate shift caused by the inputs. In such sense, it will work better than BatchNorm when the batch size is small. Besides, SphereConv is more flexible in terms of design choices (e.g. linear, cosine, and sigmoid) and each may lead to different advantages.

Similar to BatchNorm, we could use a rescaling strategy for the SphereNorm. Specifically, we rescale the output of SphereConv via $\beta\mathcal{F}_s(\boldsymbol{w}, \boldsymbol{x}) + \gamma$ where $\beta$ and $\gamma$ are learned by back-prop (similar to BatchNorm, the rescaling parameters can be either learned or prefixed). In fact, SphereNorm does not contradict with the BatchNorm at all and can be used simultaneously with BatchNorm. Interestingly, we find using both is empirically better than using either one alone.

# 3  Learning Objective on Hyperspheres

For learning on hyperspheres, we can either use the conventional loss function such as softmax loss, or use some loss functions that are tailored for the SphereConv operators. We present some possible choices for these tailored loss functions.

**Weight-normalized Softmax Loss**. The input feature and its label are denoted as $\boldsymbol{x}_i$ and $y_i$, respectively. The original softmax loss can be written as $L = \frac{1}{N}\sum_i L_i = \frac{1}{N}\sum_i -\log\left(\frac{e^{f_{y_i}}}{\sum_j e^{f_j}}\right)$ where $N$ is the number of training samples and $f_j$ is the score of the $j$-th class ($j \in [1, K]$, $K$ is the number of classes). The class score vector $\boldsymbol{f}$ is usually the output of a fully connected layer $\boldsymbol{W}$, so we have $f_j = \boldsymbol{W}_j^\top\boldsymbol{x}_i + b_j$ and $f_{y_i} = \boldsymbol{W}_{y_i}^\top\boldsymbol{x}_i + b_{y_i}$ in which $\boldsymbol{x}_i$, $\boldsymbol{W}_j$, and $\boldsymbol{W}_{y_i}$ are the $i$-th training sample, the $j$-th and $y_i$-th column of $\boldsymbol{W}$ respectively. We can rewrite $L_i$ as

$$L_i = -\log\left(\frac{e^{\boldsymbol{W}_{y_i}^\top\boldsymbol{x}_i + b_{y_i}}}{\sum_j e^{\boldsymbol{W}_j^\top\boldsymbol{x}_i + b_j}}\right) = -\log\left(\frac{e^{\|\boldsymbol{W}_{y_i}\|\|\boldsymbol{x}_i\|\cos(\theta_{y_i,i}) + b_{y_i}}}{\sum_j e^{\|\boldsymbol{W}_j\|\|\boldsymbol{x}_i\|\cos(\theta_{j,i}) + b_j}}\right), \tag{9}$$

where $\theta_{j,i}(0 \le \theta_{j,i} \le \pi)$ is the angle between vector $\boldsymbol{W}_j$ and $\boldsymbol{x}_i$. The decision boundary of the original softmax loss is determined by the vector $\boldsymbol{f}$. Specifically in the binary-class case, the

decision boundary of the softmax loss is $\boldsymbol{W}_1^\top \boldsymbol{x} + b_1 = \boldsymbol{W}_2^\top \boldsymbol{x} + b_2$. Considering the intuition of the SphereConv operators, we want to make the decision boundary only depend on the angles. To this end, we normalize the weights ($\|\boldsymbol{W}_j\| = 1$) and zero out the biases ($b_j = 0$), following the intuition in [11] (sometimes we could keep the biases while data is imbalanced). The decision boundary becomes $\|\boldsymbol{x}\| \cos(\theta_1) = \|\boldsymbol{x}\| \cos(\theta_2)$. Similar to SphereConv, we could generalize the decision boundary to $\|\boldsymbol{x}\| g(\theta_1) = \|\boldsymbol{x}\| g(\theta_2)$, so the weight-normalized softmax (W-Softmax) loss can be written as

$$L_i = -\log\left( \frac{e^{\|\boldsymbol{x}_i\| g(\theta_{y_i,i})}}{\sum_j e^{\|\boldsymbol{x}_i\| g(\theta_{j,i})}} \right), \tag{10}$$

where $g(\cdot)$ can take the form of linear SphereConv, cosine SphereConv, or sigmoid SphereConv. Thus we also term these three difference weight-normalized loss functions as linear W-Softmax loss, cosine W-Softmax loss, and sigmoid W-Softmax loss, respectively.

**Generalized Angular Softmax Loss**. Inspired by [11], we use a multiplicative parameter $m$ to impose margins on hyperspheres. We propose a generalized angular softmax (GA-Softmax) loss which extends the W-Softmax loss to a loss function that favors large angular margin feature distribution. In general, the GA-Softmax loss is formulated as

$$L_i = -\log\left( \frac{e^{\|\boldsymbol{x}_i\| g(m\theta_{y_i,i})}}{e^{\|\boldsymbol{x}_i\| g(m\theta_{y_i,i})} + \sum_{j \neq y_i} e^{\|\boldsymbol{x}_i\| g(\theta_{j,i})}} \right), \tag{11}$$

where $g(\cdot)$ could also have the linear, cosine and sigmoid form, similar to the W-Softmax loss. We can see A-Softmax loss [11] is exactly the cosine GA-Softmax loss and W-Softmax loss is the special case ($m = 1$) of GA-Sofmtax loss. Note that we usually require $\theta_{j,i} \in [0, \frac{\pi}{m}]$, because $\cos(\theta_{j,i})$ is only monotonically decreasing in $[0, \pi]$. To address this, [12, 11] construct a monotonically decreasing function recursively using the $[0, \frac{\pi}{m}]$ part of $\cos(m\theta_{j,i})$. Although it indeed partially addressed the issue, it may introduce a number of saddle points (w.r.t. $\boldsymbol{W}$) in the loss surfaces. Originally, $\frac{\partial g}{\partial \theta}$ will be close to 0 only when $\theta$ is close to 0 and $\pi$. However, in L-Softmax [12] or A-Softmax (cosine GA-Softmax), it is not the case. $\frac{\partial g}{\partial \theta}$ will be 0 when $\theta = \frac{k\theta}{m}, k = 0, \cdots, m$. It will possibly cause instability in training. The sigmoid GA-Softmax loss also has similar issues. However, if we use the linear GA-Softmax loss, this problem will be automatically solved and the training will possibly become more stable in practice. There will also be a lot of choices of $g(\cdot)$ to design a specific GA-Sofmtax loss, and each one has different optimization dynamics. The optimal one may depend on the task itself (e.g. cosine GA-Softmax has been shown effective in deep face recognition [11]).

**Discussion of Sphere-normalized Softmax Loss**. We have also considered the sphere-normalized softmax loss (S-Softmax), which simultaneously normalizes the weights ($\boldsymbol{W}_j$) and the feature $\boldsymbol{x}$. It seems to be a more natural choice than W-Softmax for the proposed SphereConv and makes the entire framework more unified. In fact, we have tried this and the empirical results are not that good, because the optimization seems to become very difficult. If we use the S-Softmax loss to train a network from scratch, we can not get reasonable results without using extra tricks, which is the reason we do not use it in this paper. For completeness, we give some discussions here. Normally, it is very difficult to make the S-Softmax loss value to be small enough, because we normalize the features to unit hypersphere. To make this loss work, we need to either normalize the feature to a value much larger than 1 (hypersphere with large radius) and then tune the learning rate or first train the network with the softmax loss from scratch and then use the S-Softmax loss for finetuning.

## 4 Experiments and Results

### 4.1 Experimental Settings

We will first perform comprehensive ablation study and exploratory experiments for the proposed SphereNets, and then evaluate the SphereNets on image classification. For the image classification task, we perform experiments on CIFAR10 (only with random left-right flipping), CIFAR10+ (with full data augmentation), CIFAR100 and large-scale Imagenet 2012 datasets [17].

**General Settings**. For CIFAR10, CIFAR10+ and CIFAR100, we follow the same settings from [7, 12]. For Imagenet 2012 dataset, we mostly follow the settings in [9]. We attach more details in Appendix B. For fairness, batch normalization and ReLU are used in all methods if not specified. All the comparisons are made to be fair. Compared CNNs have the same architecture with SphereNets.

**Training**. Appendix A gives the network details. For CIFAR-10 and CIFAR-100, we use the ADAM, starting with the learning rate 0.001. The batch size is 128 if not specified. The learning rate is divided by 10 at 34K, 54K iterations and the training stops at 64K. For both A-Softmax and GA-Softmax loss,

we use $m=4$. For Imagenet-2012, we use the SGD with momentum $0.9$. The learning rate starts with $0.1$, and is divided by 10 at 200K and 375K iterations. The training stops at 550K iteration.

## 4.2 Ablation Study and Exploratory Experiments

We perform comprehensive Ablation and exploratory study on the SphereNet and evaluate every component individually in order to analyze its advantages. We use the 9-layer CNN as default (if not specified) and perform the image classification on CIFAR-10 without any data augmentation.

| SphereConv Operator / Loss | Original Softmax | Sigmoid (0.1) W-Softmax | Sigmoid (0.3) W-Softmax | Sigmoid (0.7) W-Softmax | Linear W-Softmax | Cosine W-Softmax | A-Softmax (m=4) | GA-Softmax (m=4) |
|---|---|---|---|---|---|---|---|---|
| Sigmoid (0.1) | 90.97 | 90.91 | 90.89 | 90.88 | 91.07 | 91.13 | 91.87 | 91.99 |
| Sigmoid (0.3) | 91.08 | **91.44** | 91.37 | 91.21 | **91.34** | **91.28** | 92.13 | **92.38** |
| Sigmoid (0.7) | 91.05 | 91.16 | **91.47** | 91.07 | 90.99 | 91.18 | **92.22** | 92.36 |
| Linear | **91.10** | 90.93 | 91.42 | 90.96 | 90.95 | 91.24 | 92.21 | 92.32 |
| Cosine | 90.89 | 90.88 | 91.08 | **91.22** | 91.17 | 90.99 | 91.94 | 92.19 |
| Original Conv | 90.58 | 90.58 | 90.73 | 90.78 | 91.08 | 90.68 | 91.78 | 91.80 |

Table 1: Classification accuracy (%) with different loss functions.

**Comparison of different loss functions.** We first evaluate all the SphereConv operators with different loss functions. All the compared SphereConv operators use the 9-layer CNN architecture in the experiment. From the results in Table 1, one can observe that the SphereConv operators consistently outperforms the original convolutional operator. For the compared loss functions except A-Softmax and GA-Softmax, the effect on accuracy seems to less crucial than the SphereConv operators, but sigmoid W-Softmax is more flexible and thus works slightly better than the others. The sigmoid SphereConv operators with a suitably chosen parameter also works better than the others. Note that, W-Softmax loss is in fact comparable to the original softmax loss, because our SphereNet optimizes angles and the W-Softmax is derived from the original softmax loss. Therefore, it is fair to compare the SphereNet with W-Softmax and CNN with softmax loss. From Table 1, we can see SphereConv operators are consistently better than the covolutional operators. While we use a large-margin loss function like the A-Softmax [11] and the proposed GA-Softmax, the accuracy can be further boosted. One may notice that A-Softmax is actually cosine GA-Softmax. The superior performance of A-Softmax with SphereNet shows that our architecture is more suitable for the learning of angular loss. Moreover, our proposed large-margin loss (linear GA-Softmax) performs the best among all these compared loss functions.

**Comparison of different network architectures.** We are also interested in how our SphereConv operators work in different architectures. We evaluate all the proposed SphereConv operators with the same architecture of different layers and a totally different architecture (ResNet). Our baseline CNN architecture follows the design of VGG network [18] only with different convolutional layers. For fair comparison, we use cosine W-Softmax for all SphereConv operators and original softmax for original convolution operators. From the results in Table 2, one can see that SphereNets greatly outperforms the CNN baselines, usually with more than 1% improvement. While applied to ResNet, our SphereConv operators also work better than the baseline. Note that, we use the similar ResNet architecture from the CIFAR-10 experiment in [6]. We do not use data augmentation for CIFAR-10 in this experiment, so the ResNet accuracy is much lower than the reported one in [6]. Our results on different network architectures show consistent and significant improvement over CNNs.

| SphereConv Operator | CNN-3 | CNN-9 | CNN-18 | CNN-45 | CNN-60 | ResNet-32 |
|---|---|---|---|---|---|---|
| Sigmoid (0.1) | 82.08 | 91.13 | 91.43 | 89.34 | 87.67 | 90.94 |
| Sigmoid (0.3) | 81.92 | **91.28** | 91.55 | 89.73 | 87.85 | **91.7** |
| Sigmoid (0.7) | 82.4 | 91.18 | **91.69** | 89.85 | 88.42 | 91.19 |
| Linear | **82.31** | 91.15 | 91.24 | **90.15** | **89.91** | 91.25 |
| Cosine | 82.23 | 90.99 | 91.23 | 90.05 | 89.28 | 91.38 |
| Original Conv | 81.19 | 90.68 | 90.62 | 88.23 | 88.15 | 90.40 |

| SphereConv Operator | Acc. (%) |
|---|---|
| Sigmoid (0.1) | **86.29** |
| Sigmoid (0.3) | 85.67 |
| Sigmoid (0.7) | 85.51 |
| Linear | 85.34 |
| Cosine | 85.25 |
| CNN w/o ReLU | 80.73 |

Table 2: Classification accuracy (%) with different network architectures.    Table 3: Acc. w/o ReLU.

**Comparison of different width (number of filters).** We evaluate the SphereNet with different number of filters. Fig. 3(c) shows the convergence of different width of SphereNets. 16/32/48 means conv1.x, conv2.x and conv3.x have 16, 32 and 48 filters, respectively. One could observe that while the number of filters are small, SphereNet performs similarly to CNNs (slightly worse). However, while we increase the number of filters, the final accuracy will surpass the CNN baseline even faster and more stable convergence performance. With large width, we find that SphereNets perform consistently better than CNN baselines, showing that SphereNets can make better use of the width.

**Learning without ReLU.** We notice that SphereConv operators are no longer a matrix multiplication, so it is essentially a non-linear function. Because the SphereConv operators already introduce certain

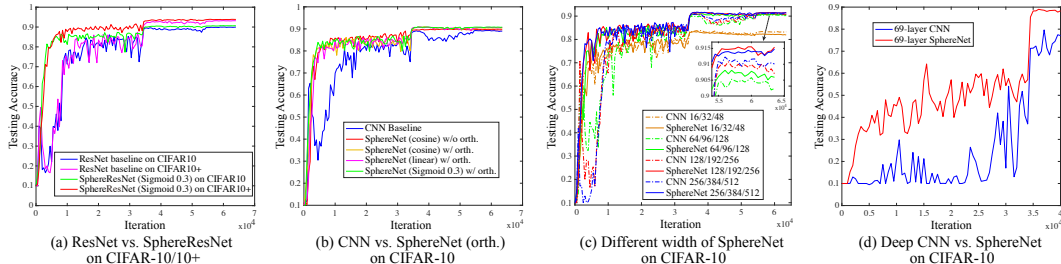

(a) ResNet vs. SphereResNet on CIFAR-10/10+     (b) CNN vs. SphereNet (orth.) on CIFAR-10     (c) Different width of SphereNet on CIFAR-10     (d) Deep CNN vs. SphereNet on CIFAR-10

Figure 3: Testing accuracy over iterations. (a) ResNet vs. SphereResNet. (b) Plain CNN vs. plain SphereNet. (c) Different width of SphereNet. (d) Ultra-deep plain CNN vs. ultra-deep plain SphereNet.

non-linearity to the network, we evaluate how much gain will such non-linearity bring. Therefore, we remove the ReLU activation and compare our SphereNet with the CNNs without ReLU. The results are given in Table 3. All the compared methods use 18-layer CNNs (with BatchNorm). Although removing ReLU greatly reduces the classification accuracy, our SphereNet still outperforms the CNN without ReLU by a significant margin, showing its rich non-linearity and representation power.

**Convergence.** One of the most significant advantages of SphereNet is its training stability and convergence speed. We evaluate the convergence with two different architectures: CNN-9 and ResNet-32. For fair comparison, we use the original softmax loss for all compared methods (including SphereNets). ADAM is used for the stochastic optimization and the learning rate is the same for all networks. From Fig. 3(a), the SphereResNet converges significantly faster than the original ResNet baseline in both CIFAR-10 and CIFAR-10+ and the final accuracy are also higher than the baselines. In Fig. 3(b), we evaluate the SphereNet with and without orthogonality constraints on kernel weights. With the same network architecture, SphereNet also converges much faster and performs better than the baselines. The orthogonality constraints also can bring performance gains in some cases. Generally from Fig. 3, one could also observe that the SphereNet converges fast and very stably in every case while the CNN baseline fluctuates in a relative wide range.

**Optimizing ultra-deep networks.** Partially because of the alleviation of the covariate shift problem and the improvement of conditioning, our SphereNet is able to optimize ultra-deep neural networks without using residual units or any form of shortcuts. For SphereNets, we use the cosine SphereConv operator with the cosine W-Softmax loss. We directly optimize a very deep plain network with 69 stacked convolutional layers. From Fig. 3(d), one can see that the convergence of SphereNet is much easier than the CNN baseline and the SphereNet is able to achieve nearly 90% final accuracy.

### 4.3 Preliminary Study towards Learnable SphereConv

Although the learnable SphereConv is not a main theme of this paper, we still run some preliminary evaluations on it. For the proposed learnable sigmoid SphereConv, we learn the parameter $k$ independently for each filter. It is also trivial to learn it in a layer-shared or network-shared fashsion. With the same 9-layer architecture used in Section 4.2, the learnable SphereConv (with cosine W-Softmax loss) achieves 91.64% on CIFAR-10 (without full data augmentation), while the best sigmoid SphereConv (with cosine W-Softmax loss) achieves 91.22%. In Fig. 4, we also plot the frequency histogram of $k$ in Conv1.1 (64 filters), Conv2.1 (96 filters) and Conv3.1 (128 filters) of the final learned SphereNet.

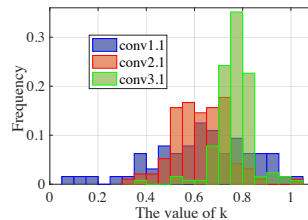

Figure 4: Frequency histogram of $k$.

From Fig. 4, we observe that each layer learns different distribution of $k$. The first convolutional layer (Conv1.1) tends to uniformly distribute $k$ into a large range of values from 0 to 1, potentially extracting information from all levels of angular similarity. The fourth convolutional layer (Conv2.1) tends to learn more concentrated distribution of $k$ than Conv1.1, while the seventh convolutional layer (Conv3.1) learns highly concentrated distribution of $k$ which is centered around 0.8. Note that, we initialize all $k$ with a constant 0.5 and learn them with the back-prop.

### 4.4 Evaluation of SphereNorm

From Section 4.2, we could clearly see the convergence advantage of SphereNets. In general, we can view the SphereConv as a normalization method (comparable to batch normalization) that can be applied to all kinds of networks. This section evaluates the challenging scenarios where the mini-batch size is small (results under 128 batch size could be found in Section 4.2) and we use the same

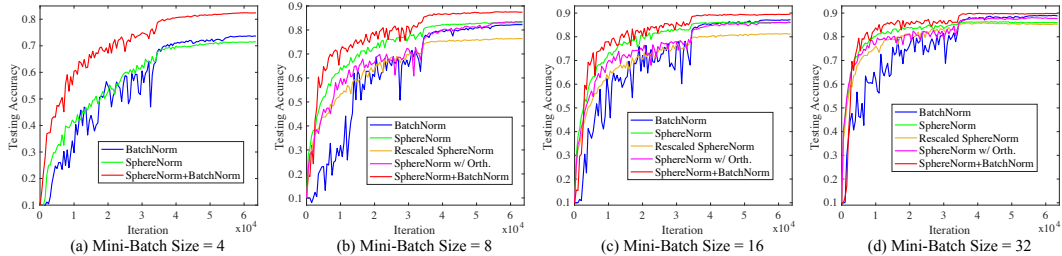

Figure 5: Convergence under different mini-batch size on CIFAR-10 dataset (Same setting as Section 4.2).

9-layer CNN as in Section 4.2. To be simple, we use the cosine SphereConv as SphereNorm. The softmax loss is used in both CNNs and SphereNets. From Fig. 5, we could observe that SphereNorm achieves the final accuracy similar to BatchNorm, but SphereNorm converges faster and more stably. SphereNorm plus the orthogonal constraint helps convergence a little bit and rescaled SphereNorm does not seem to work well. While BatchNorm and SphereNorm are used together, we obtain the fastest convergence and the highest final accuracy, showing excellent compatibility of SphereNorm.

### 4.5 Image Classification on CIFAR-10+ and CIFAR-100

We first evaluate the SphereNet in a classic image classification task. We use the CIFAR-10+ and CIFAR100 datasets and perform random flip (both horizontal and vertical) and random crop as data augmentation (CIFAR-10 with full data augmentation is denoted as CIFAR-10+). We use the ResNet-32 as a baseline architecture. For the SphereNet of the same architecture, we evaluate sigmoid SphereConv operator ($k = 0.3$) with sigmoid W-Softmax ($k = 0.3$) loss (S-SW), linear SphereConv operator with linear W-Softmax loss

| Method | CIFAR-10+ | CIFAR-100 |
|---|---|---|
| ELU [2] | 94.16 | 72.34 |
| FitResNet (LSUV) [14] | 93.45 | 65.72 |
| ResNet-1001 [7] | **95.38** | **77.29** |
| Baseline ResNet-32 (softmax) | 93.26 | 72.85 |
| SphereResNet-32 (S-SW) | 94.47 | 76.02 |
| SphereResNet-32 (L-LW) | 94.33 | 75.62 |
| SphereResNet-32 (C-CW) | 94.64 | 74.92 |
| SphereResNet-32 (S-G) | **95.01** | **76.39** |

Table 4: Acc. (%) on CIFAR-10+ & CIFAR-100.

(L-LW), cosine SphereConv operator with cosine W-Softmax loss (C-CW) and sigmoid SphereConv operator ($k = 0.3$) with GA-Softmax loss (S-G). In Table 4, we could see the SphereNet outperforms a lot of current state-of-the-art methods and is even comparable to the ResNet-1001 which is far deeper than ours. This experiment further validates our idea that learning on a hyperspheres constrains the parameter space to a more semantic and label-related one.

### 4.6 Large-scale Image Classification on Imagenet-2012

We evaluate SphereNets on large-scale Imagenet-2012 dataset. We only use the minimum data augmentation strategy in the experiment (details are in Appendix B). For the ResNet-18 baseline and SphereResNet-18, we use the same filter numbers in each layer. We develop two types of SphereResNet-18, termed as v1 and v2 respectively. In SphereResNet-18-v2, we do not use SphereConv in the $1 \times 1$ shortcut convolutions which are used to match the number of channels. In SphereResNet-18-v1, we use SphereConv in

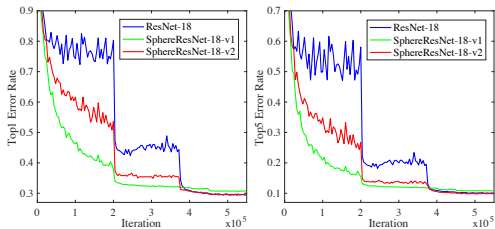

Figure 6: Validation error (%) on ImageNet.

the $1 \times 1$ shortcut convolutions. Fig. 6 shows the single crop validation error over iterations. One could observe that both SphereResNets converge much faster than the ResNet baseline, while SphereResNet-18-v1 converges the fastest but yields a slightly worse yet comparable accuracy. SphereResNet-18-v2 not only converges faster than ResNet-18, but it also shows slightly better accuracy.

## 5 Limitations and Future Work

Our work still has some limitations: (1) SphereNets have large performance gain while the network is wide enough. If the network is not wide enough, SphereNets still converge much faster but yield slightly worse (still comparable) recognition accuracy. (2) The computation complexity of each neuron is slightly higher than the CNNs. (3) SphereConvs are still mostly prefixed. Possible future work includes designing/learning a better SphereConv, efficiently computing the angles to reduce computation complexity, applications to the tasks that require fast convergence (e.g. reinforcement learning and recurrent neural networks), better angular regularization to replace orthogonality, etc.

## Acknowledgements

We thank Zhen Liu (Georgia Tech) for helping with the experiments and providing suggestions. This project was supported in part by NSF IIS-1218749, NIH BIGDATA 1R01GM108341, NSF CAREER IIS-1350983, NSF IIS-1639792 EAGER, NSF CNS-1704701, ONR N00014-15-1-2340, Intel ISTC, NVIDIA and Amazon AWS. Xingguo Li is supported by doctoral dissertation fellowship from University of Minnesota. Yan-Ming Zhang is supported by the National Natural Science Foundation of China under Grant 61773376.

## Footnotes

[1]Without loss of generality, we study CNNs here, but our method is generalizable to any other neural nets.

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
