[Supplementary Material]

## A  Network Architectures

| Layer | CNN-3 | CNN-9 | CNN-18 | CNN-45 | CNN-60 | CNN-69 |
|---|---|---|---|---|---|---|
| Conv1.x | [3×3, 64]×1 | [3×3, 64]×3 | [3×3, 64]×6 | [3×3, 64]×15 | [3×3, 64]×20 | [3×3, 64]×23 |
| Pool1 | 2×2 Max Pooling, Stride 2 | | | | | |
| Conv2.x | [3×3, 96]×1 | [3×3, 96]×3 | [3×3, 96]×6 | [3×3, 96]×15 | [3×3, 96]×20 | [3×3, 96]×23 |
| Pool2 | 2×2 Max Pooling, Stride 2 | | | | | |
| Conv3.x | [3×3, 128]×1 | [3×3, 128]×3 | [3×3, 128]×6 | [3×3, 128]×15 | [3×3, 128]×20 | [3×3, 128]×23 |
| Pool3 | 2×2 Max Pooling, Stride 2 | | | | | |
| Fully Connected | 256 | 256 | 256 | 256 | 256 | 256 |

Table 5: Our plain CNN architectures with different convolutional layers. Conv1.x, Conv2.x and Conv3.x denote convolution units that may contain multiple convolution layers. E.g., [3×3, 64]×3 denotes 3 cascaded convolution layers with 64 filters of size 3×3.

| Layer | ResNet-32 for Section 4.2 | ResNet-32 for Section 4.3 | ResNet-18 for Section 4.6 |
|---|---|---|---|
| Conv0.x | N/A | N/A | [7×7, 256], Stride 2 <br> 3×3, Max Pooling, Stride 2 |
| Conv1.x | [3×3, 64]×1 <br> $\begin{bmatrix} 3\times3, 64 \\ 3\times3, 64 \end{bmatrix} \times 5$ | [3×3, 96]×1 <br> $\begin{bmatrix} 3\times3, 96 \\ 3\times3, 96 \end{bmatrix} \times 5$ | $\begin{bmatrix} 3\times3, 256 \\ 3\times3, 256 \end{bmatrix} \times 2$ |
| Conv2.x | $\begin{bmatrix} 3\times3, 96 \\ 3\times3, 96 \end{bmatrix} \times 5$ | $\begin{bmatrix} 3\times3, 192 \\ 3\times3, 192 \end{bmatrix} \times 5$ | $\begin{bmatrix} 3\times3, 512 \\ 3\times3, 512 \end{bmatrix} \times 2$ |
| Conv3.x | $\begin{bmatrix} 3\times3, 128 \\ 3\times3, 128 \end{bmatrix} \times 5$ | $\begin{bmatrix} 3\times3, 384 \\ 3\times3, 384 \end{bmatrix} \times 5$ | $\begin{bmatrix} 3\times3, 768 \\ 3\times3, 768 \end{bmatrix} \times 2$ |
| Conv4.x | N/A | N/A | $\begin{bmatrix} 3\times3, 1024 \\ 3\times3, 1024 \end{bmatrix} \times 2$ |
| | Average Pooling | | |

Table 6: Our ResNet architectures with different convolutional layers. Conv0.x, Conv1.x, Conv2.x, Conv3.x and Conv4.x denote convolution units that may contain multiple convolutional layers, and residual units are shown in double-column brackets. Conv1.x, Conv2.x and Conv3.x usually operate on different size feature maps. These networks are essentially the same as [6], but some may have different number of filters in each layer. The downsampling is performed by convolutions with a stride of 2. E.g., [3×3, 64]×4 denotes 4 cascaded convolution layers with 64 filters of size 3×3, and S2 denotes stride 2.

## B  Experimental Details for Imagenet-2012

For the input data of the Imagenet-2012 experiment, we only use the minimum data augmentation. Specifically, we first resize the images to $256 \times 256$ resolution and then randomly crop patches of size $224 \times 224$ from the resized images. Besides that, we also randomly flip the image horizontally. For SphereResNet-18-v1, we use the cosine SphereConv and the cosine W-Softmax loss. For SphereResNet-18-v2, we use the cosine SphereConv and the softmax loss. Generally, we find that the standard softmax loss and all kinds of W-Softmax loss usually have similar empirical performance. Note that, we could obtain better performance by using the other SphereConvs (sigmoid SphereConv with $k = 0.3$ is a good choice), but it requires more GPU memory. Due to the width of our architecture and the limitation of GPU memory, the mini-batch size is set to $40$ for all methods in the Imagenet-2012 experiment.

## C  More Discussions for Sphere-normalized Softmax Loss

The sphere-normalized softmax (S-Softmax) loss is essentially applying the SphereConv to the fully connected layer in the softmax loss[2]. However, simply applying the SphereConv can not make such loss work, because this loss function is difficult to converge in practice. To address this, we rescale the logit output of the S-Softmax loss with a scaling factor $s$. Therefore, the output range is changed from $[-1, 1]$ to $[-s, s]$. Typically, setting $s$ from 10 to 70 works pretty well in practice. We could also use the cross-validation strategy to set the hyperparameter $s$.

# D Proofs of Lemmas

## D.1 Proof of Lemma 1

The gradient is

$$\nabla\mathcal{G}(\boldsymbol{U},\boldsymbol{V}) = \left[ \begin{array}{c} \nabla_{\boldsymbol{U}}\mathcal{G}(\boldsymbol{U},\boldsymbol{V}) \\ \nabla_{\boldsymbol{V}}\mathcal{G}(\boldsymbol{U},\boldsymbol{V}) \end{array} \right] = \left[ \begin{array}{c} (\boldsymbol{U}\boldsymbol{V}^\top - \boldsymbol{F})\boldsymbol{V} \\ (\boldsymbol{V}\boldsymbol{U}^\top - \boldsymbol{F}^\top)\boldsymbol{U} \end{array} \right]$$

The Hessian matrix is

$$\begin{aligned}
\nabla^2\mathcal{G}(\boldsymbol{U},\boldsymbol{V}) &= \left[ \begin{array}{cc} \nabla_{\boldsymbol{U}}^2\mathcal{G}(\boldsymbol{U},\boldsymbol{V}) & \nabla_{\boldsymbol{U},\boldsymbol{V}}^2\mathcal{G}(\boldsymbol{U},\boldsymbol{V}) \\ \nabla_{\boldsymbol{V},\boldsymbol{U}}^2\mathcal{G}(\boldsymbol{U},\boldsymbol{V}) & \nabla_{\boldsymbol{V}}^2\mathcal{G}(\boldsymbol{U},\boldsymbol{V}) \end{array} \right] \\
&= \left[ \begin{array}{cc} \boldsymbol{V}^\top\boldsymbol{V} \otimes \boldsymbol{I}_n & (\boldsymbol{U}\boldsymbol{V}^\top - \boldsymbol{F}) \otimes \boldsymbol{I}_k + \boldsymbol{U} \boxtimes \boldsymbol{V} \\ (\boldsymbol{V}\boldsymbol{U}^\top - \boldsymbol{F}^\top) \otimes \boldsymbol{I}_k + \boldsymbol{V} \boxtimes \boldsymbol{U} & \boldsymbol{U}^\top\boldsymbol{U} \otimes \boldsymbol{I}_m \end{array} \right],
\end{aligned} \qquad (12)$$

where $\boldsymbol{I}_n$ is an $n \times n$ identity matrix for any integer $n$, given matrices $\boldsymbol{A} \in \mathbb{R}^{n \times r}$ and $\boldsymbol{B} \in \mathbb{R}^{m \times k}$ with $\boldsymbol{A}_{:,i}$ denoting the $i$-th column of $\boldsymbol{A}$, $\boldsymbol{A} \boxtimes \boldsymbol{B} \in \mathbb{R}^{nk \times mr}$ is defined as

$$\boldsymbol{A} \boxtimes \boldsymbol{B} = \left[ \begin{array}{cccc} \boldsymbol{A}_{:,1}\boldsymbol{B}_{:,1}^\top & \boldsymbol{A}_{:,2}\boldsymbol{B}_{:,1}^\top & \cdots & \boldsymbol{A}_{:,r}\boldsymbol{B}_{:,1}^\top \\ \boldsymbol{A}_{:,1}\boldsymbol{B}_{:,2}^\top & \boldsymbol{A}_{:,2}\boldsymbol{B}_{:,2}^\top & \cdots & \boldsymbol{A}_{:,r}\boldsymbol{B}_{:,2}^\top \\ \vdots & \vdots & \ddots & \vdots \\ \boldsymbol{A}_{:,1}\boldsymbol{B}_{:,k}^\top & \boldsymbol{A}_{:,2}\boldsymbol{B}_{:,k}^\top & \cdots & \boldsymbol{A}_{:,r}\boldsymbol{B}_{:,k}^\top \end{array} \right].$$

At a global optimum, we have $\boldsymbol{U}\boldsymbol{V}^\top = \boldsymbol{F}$. Then it is easy to see that for any real $c$, if $\widetilde{\boldsymbol{U}} = c\boldsymbol{U}$ and $\widetilde{\boldsymbol{V}} = \boldsymbol{V}/c$, then we have

$$\nabla^2\mathcal{G}(\widetilde{\boldsymbol{U}},\widetilde{\boldsymbol{V}}) = \left[ \begin{array}{cc} \frac{1}{c^2}\boldsymbol{V}^\top\boldsymbol{V} \otimes \boldsymbol{I}_n & \boldsymbol{U} \boxtimes \boldsymbol{V} \\ \boldsymbol{V} \boxtimes \boldsymbol{U} & c^2\boldsymbol{U}^\top\boldsymbol{U} \otimes \boldsymbol{I}_m \end{array} \right]$$

We have that at a global optimal point, $\nabla^2\mathcal{G}(\boldsymbol{U},\boldsymbol{V})$ is a positive semidefinite matrix with the smallest eigenvalue equal to 0. Specifically, due to the existence of the invariance, i.e., $\boldsymbol{U}\boldsymbol{V}^\top = \boldsymbol{U}\boldsymbol{R}(\boldsymbol{V}\boldsymbol{R})^\top$ for any orthogonal matrix $\boldsymbol{R} \in \mathbb{R}^{r \times r}$, there are $r(r-1)/2$ number of eigenvectors of $\nabla^2\mathcal{G}(\boldsymbol{U},\boldsymbol{V})$ at $\boldsymbol{U}\boldsymbol{V}^\top = \boldsymbol{F}$ corresponding to 0 eigenvalue [10]. Then for any $c > 1$, we have

$$\begin{aligned}
\mathrm{Tr}(\nabla^2\mathcal{G}(\widetilde{\boldsymbol{U}},\widetilde{\boldsymbol{V}})) &= \frac{1}{c^2}\mathrm{Tr}(\boldsymbol{V}^\top\boldsymbol{V} \otimes \boldsymbol{I}_n) + c^2\mathrm{Tr}(\boldsymbol{U}^\top\boldsymbol{U} \otimes \boldsymbol{I}_m) \\
&\geq \frac{c^2}{2}\left(\mathrm{Tr}(\boldsymbol{V}^\top\boldsymbol{V} \otimes \boldsymbol{I}_n) + \mathrm{Tr}(\boldsymbol{U}^\top\boldsymbol{U} \otimes \boldsymbol{I}_m)\right) = \frac{c^2}{2}\mathrm{Tr}(\nabla^2\mathcal{G}(\boldsymbol{U},\boldsymbol{V})).
\end{aligned}$$

This indicates that the largest eigenvalue of $\nabla^2\mathcal{G}(\widetilde{\boldsymbol{U}},\widetilde{\boldsymbol{V}})$ is on the order of $\Theta(c^2)$ times the largest eigenvalue of $\nabla^2\mathcal{G}(\boldsymbol{U},\boldsymbol{V})$ following the perturbation bound analysis [15] and $\boldsymbol{U}$ and $\boldsymbol{V}$ are balanced. Using a similar idea, the smallest nonzero eigenvalue of $\nabla^2\mathcal{G}(\widetilde{\boldsymbol{U}},\widetilde{\boldsymbol{V}})$ is no greater than the smallest nonzero eigenvalue of $\nabla^2\mathcal{G}(\boldsymbol{U},\boldsymbol{V})$, which results in our claim on the restricted condition number.

## D.2 Proof of Lemma 2

The gradient of $\mathcal{G}_S(\boldsymbol{U},\boldsymbol{V})$ is

$$\nabla\mathcal{G}_S(\boldsymbol{U},\boldsymbol{V}) = \left[ \begin{array}{c} \nabla_{\boldsymbol{U}}\mathcal{G}_S(\boldsymbol{U},\boldsymbol{V}) \\ \nabla_{\boldsymbol{V}}\mathcal{G}_S(\boldsymbol{U},\boldsymbol{V}) \end{array} \right] \quad \text{with}$$

$$\nabla_{\boldsymbol{U}}\mathcal{G}_S(\boldsymbol{U},\boldsymbol{V}) = \boldsymbol{D}_{\boldsymbol{U}}(\boldsymbol{D}_{\boldsymbol{U}}\boldsymbol{U}\boldsymbol{V}^\top\boldsymbol{D}_{\boldsymbol{V}} - \boldsymbol{F})\boldsymbol{D}_{\boldsymbol{V}}\boldsymbol{V} - \left(\boldsymbol{D}_{\boldsymbol{U}}^3(\boldsymbol{D}_{\boldsymbol{U}}\boldsymbol{U}\boldsymbol{V}^\top\boldsymbol{D}_{\boldsymbol{V}} - \boldsymbol{F}) \circledast_k (\boldsymbol{U}\boldsymbol{V}^\top\boldsymbol{D}_{\boldsymbol{V}})\right) \odot \boldsymbol{U},$$

$$\nabla_{\boldsymbol{V}}\mathcal{G}_S(\boldsymbol{U},\boldsymbol{V}) = \boldsymbol{D}_{\boldsymbol{V}}(\boldsymbol{D}_{\boldsymbol{V}}\boldsymbol{V}\boldsymbol{U}^\top\boldsymbol{D}_{\boldsymbol{U}} - \boldsymbol{F}^\top)\boldsymbol{D}_{\boldsymbol{U}}\boldsymbol{U} - \left(\boldsymbol{D}_{\boldsymbol{V}}^3(\boldsymbol{D}_{\boldsymbol{V}}\boldsymbol{V}\boldsymbol{U}^\top\boldsymbol{D}_{\boldsymbol{U}} - \boldsymbol{F}^\top) \circledast_k (\boldsymbol{V}\boldsymbol{U}^\top\boldsymbol{D}_{\boldsymbol{U}})\right) \odot \boldsymbol{V},$$

Note that after each iteration of SGD, we perform the entry-wise normalization for both $\boldsymbol{U}$ and $\boldsymbol{V}$, which means $\boldsymbol{D}_{\boldsymbol{U}} = \boldsymbol{I}_n$ and $\boldsymbol{D}_{\boldsymbol{V}} = \boldsymbol{I}_m$. Then the gradient of $\mathcal{G}_S(\boldsymbol{U},\boldsymbol{V})$ is

$$\nabla\mathcal{G}_S(\boldsymbol{U},\boldsymbol{V}) = \left[ \begin{array}{c} \nabla_{\boldsymbol{U}}\mathcal{G}_S(\boldsymbol{U},\boldsymbol{V}) \\ \nabla_{\boldsymbol{V}}\mathcal{G}_S(\boldsymbol{U},\boldsymbol{V}) \end{array} \right] = \left[ \begin{array}{c} (\boldsymbol{U}\boldsymbol{V}^\top - \boldsymbol{F})\boldsymbol{V} - \left((\boldsymbol{U}\boldsymbol{V}^\top - \boldsymbol{F}) \circledast_k (\boldsymbol{U}\boldsymbol{V}^\top)\right) \odot \boldsymbol{U} \\ (\boldsymbol{V}\boldsymbol{U}^\top - \boldsymbol{F}^\top)\boldsymbol{U} - \left((\boldsymbol{V}\boldsymbol{U}^\top - \boldsymbol{F}^\top) \circledast_k (\boldsymbol{V}\boldsymbol{U}^\top)\right) \odot \boldsymbol{V} \end{array} \right],$$

where given matrices $\boldsymbol{A}, \boldsymbol{B} \in \mathbb{R}^{n \times m}$ with $\boldsymbol{A}_{:,i}$ denoting the $i$-th column of $\boldsymbol{A}$, $\boldsymbol{A} \odot \boldsymbol{B} \in \mathbb{R}^{n \times m}$ is the Hadamard (pointwise) product, and the operation $\boldsymbol{A} \circledast_k \boldsymbol{B} \in \mathbb{R}^{n \times k}$ is defined as

$$\boldsymbol{A} \circledast_k \boldsymbol{B} = \left[ \begin{array}{c} \boldsymbol{A}_{1,:} \boldsymbol{B}_{1,:}^\top \\ \boldsymbol{A}_{2,:} \boldsymbol{B}_{2,:}^\top \\ \vdots \\ \boldsymbol{A}_{n,:} \boldsymbol{B}_{n,:}^\top \end{array} \right] \mathbf{1}_{1 \times k},$$

where $\mathbf{1}_{1 \times k}$ is a $1 \times k$ vector with all entries equal to 1.

Consequently, the Hessian matrix is

$$\nabla^2 \mathcal{G}_S(\boldsymbol{U}, \boldsymbol{V}) = \left[ \begin{array}{cc} \nabla_{\boldsymbol{U}}^2 \mathcal{G}_S(\boldsymbol{U}, \boldsymbol{V}) & \nabla_{\boldsymbol{U}, \boldsymbol{V}}^2 \mathcal{G}_S(\boldsymbol{U}, \boldsymbol{V}) \\ \nabla_{\boldsymbol{V}, \boldsymbol{U}}^2 \mathcal{G}_S(\boldsymbol{U}, \boldsymbol{V}) & \nabla_{\boldsymbol{V}}^2 \mathcal{G}_S(\boldsymbol{U}, \boldsymbol{V}) \end{array} \right] \text{ with}$$

$$\nabla_{\boldsymbol{U}}^2 \mathcal{G}_S(\boldsymbol{U}, \boldsymbol{V}) = \boldsymbol{V}^\top \boldsymbol{V} \otimes \boldsymbol{I}_n - \operatorname{diag}\left( \operatorname{vec}\left( (\boldsymbol{U}\boldsymbol{V}^\top - \boldsymbol{F}) \circledast_k (\boldsymbol{U}\boldsymbol{V}^\top) \right) \right)$$
$$- \left[ \begin{array}{ccc} \operatorname{diag}\left( \boldsymbol{U}_{:,1} \odot \left( (2\boldsymbol{U}\boldsymbol{V}^\top - \boldsymbol{F})\boldsymbol{V}_{:,1} \right) \right) & \cdots & \operatorname{diag}\left( \boldsymbol{U}_{:,1} \odot \left( (2\boldsymbol{U}\boldsymbol{V}^\top - \boldsymbol{F})\boldsymbol{V}_{:,k} \right) \right) \\ \vdots & \ddots & \vdots \\ \operatorname{diag}\left( \boldsymbol{U}_{:,k} \odot \left( (2\boldsymbol{U}\boldsymbol{V}^\top - \boldsymbol{F})\boldsymbol{V}_{:,1} \right) \right) & \cdots & \operatorname{diag}\left( \boldsymbol{U}_{:,k} \odot \left( (2\boldsymbol{U}\boldsymbol{V}^\top - \boldsymbol{F})\boldsymbol{V}_{:,k} \right) \right) \end{array} \right]$$

$$\nabla_{\boldsymbol{U}, \boldsymbol{V}}^2 \mathcal{G}_S(\boldsymbol{U}, \boldsymbol{V}) = \boldsymbol{I}_k \otimes (\boldsymbol{U}\boldsymbol{V}^\top - \boldsymbol{F}) + \boldsymbol{U} \boxtimes \boldsymbol{V}$$
$$- \left[ \begin{array}{ccc} (2\boldsymbol{U}\boldsymbol{V}^\top - \boldsymbol{F}) \odot \left( (\boldsymbol{U}_{:,1} \odot \boldsymbol{U}_{:1})\mathbf{1}_{1 \times m} \right) & \cdots & (2\boldsymbol{U}\boldsymbol{V}^\top - \boldsymbol{F}) \odot \left( (\boldsymbol{U}_{:,1} \odot \boldsymbol{U}_{:k})\mathbf{1}_{1 \times m} \right) \\ \vdots & \ddots & \vdots \\ (2\boldsymbol{U}\boldsymbol{V}^\top - \boldsymbol{F}) \odot \left( (\boldsymbol{U}_{:,k} \odot \boldsymbol{U}_{:1})\mathbf{1}_{1 \times m} \right) & \cdots & (2\boldsymbol{U}\boldsymbol{V}^\top - \boldsymbol{F}) \odot \left( (\boldsymbol{U}_{:,k} \odot \boldsymbol{U}_{:k})\mathbf{1}_{1 \times m} \right) \end{array} \right]$$

$$\nabla_{\boldsymbol{V}, \boldsymbol{U}}^2 \mathcal{G}_S(\boldsymbol{U}, \boldsymbol{V}) = \left( \nabla_{\boldsymbol{U}, \boldsymbol{V}}^2 \mathcal{G}_S(\boldsymbol{U}, \boldsymbol{V}) \right)^\top$$

$$\nabla_{\boldsymbol{V}}^2 \mathcal{G}_S(\boldsymbol{U}, \boldsymbol{V}) = \boldsymbol{U}^\top \boldsymbol{U} \otimes \boldsymbol{I}_n - \operatorname{diag}\left( \operatorname{vec}\left( (\boldsymbol{V}\boldsymbol{U}^\top - \boldsymbol{F}^\top) \circledast_k (\boldsymbol{V}\boldsymbol{U}^\top) \right) \right)$$
$$- \left[ \begin{array}{ccc} \operatorname{diag}\left( \boldsymbol{V}_{:,1} \odot \left( (2\boldsymbol{V}\boldsymbol{U}^\top - \boldsymbol{F}^\top)\boldsymbol{U}_{:,1} \right) \right) & \cdots & \operatorname{diag}\left( \boldsymbol{V}_{:,1} \odot \left( (2\boldsymbol{V}\boldsymbol{U}^\top - \boldsymbol{F}^\top)\boldsymbol{U}_{:,k} \right) \right) \\ \vdots & \ddots & \vdots \\ \operatorname{diag}\left( \boldsymbol{V}_{:,k} \odot \left( (2\boldsymbol{V}\boldsymbol{U}^\top - \boldsymbol{F}^\top)\boldsymbol{U}_{:,1} \right) \right) & \cdots & \operatorname{diag}\left( \boldsymbol{V}_{:,k} \odot \left( (2\boldsymbol{V}\boldsymbol{U}^\top - \boldsymbol{F}^\top)\boldsymbol{U}_{:,k} \right) \right) \end{array} \right]$$

Then we have $\lambda_i(\nabla^2 \mathcal{G}_S(\widetilde{\boldsymbol{U}}, \widetilde{\boldsymbol{V}})) = \lambda_i(\nabla^2 \mathcal{G}_S(\boldsymbol{U}, \boldsymbol{V}))$ for all $i \in [(n+m)k] = \{1, 2, \ldots, (n+m)k\}$ by noticing that we normalize the data as $\frac{\boldsymbol{U}_{i,j}}{\|\boldsymbol{U}_{i,:}\|_2}$ for all $i \in [n]$ and $\frac{\boldsymbol{V}_{i,j}}{\|\boldsymbol{V}_{i,:}\|_2}$ for all $i \in [m]$. This finishes the proof.

## Footnotes

[2]The softmax loss is defined as the combination of the last fully connected layer, the softmax function and the cross-entropy loss.