[Reviews · NeurIPS 2017]

Reviewer 1



This paper presents a very neat idea of using a unique operator to replace the original conv filter operation, which is matrix inner product. Instead of computing similarity between filter weights and pixel values (or activation values if intermediate layers), they compute the angle between them in the hypersphere space. They also defined three types of operators all as functions of the angle: linear, cosine, and sigmoid. To better suit such an operation they also redefined the regularization term and loss function. Ablation study shows the effectiveness of each addition introduced, and better results are generated for image classification task in CIFAR 10 and CIFAR 100 and image feature embedding task. Improvements are seen in terms of both higher accuracy and easier and faster convergence. The paper is theoretically sound, clearly structured, experiments well organized, and the writing and presentation is elegant. The drawback, though, is that the author didn't extend experiments to larger and more realistic image classification tasks such as ImageNet, and gave no explanation or hint of future work on it. It can also be worthwhile to test with more diverse tasks such as certain small scale reinforcement learning problems.

Reviewer 2



This paper presents a novel architecture, SphereNet, which replaces the traditional dot product with geodesic distance as the convolution operators and fully-connected layers. SphereNet also regularizes the weights for softmax to be norm 1 for angular softmax. The results show that SphereNet can achieve superior performance in terms of accuracy and convergence rate as well as mitigating the vanishing/exploding gradients in deep networks. Novelty: Replacing dot product similarity with angular similarity has widely existed in the deep learning literature. With that being said, most works focus on using angular similarity for Softmax or loss functions. This paper introduces spherical operation for convolution, which is novel in the literature as far as I know. Significance: The spherical operations the paper introduces achieve faster convergence rate and better accuracy performance. It also mitigates the vanishing/exploding gradients brought by dot products, which is a long standing problem. It will be interesting to lots of people in ML community. Improvement: I have several suggestions and questions as listed below: - For angular Softmax, the bias is removed from the equation. However, the existence of bias is very important for calibrating the output, especially for imbalanced datasets. It would be great if the authors could add that back. Otherwise, it's better to try current model on an imbalanced dataset to measure the effect. - In the first two subfigures of Figure 4, the baseline methods such as standard CNN actually converges faster at the very beginning. But there is a big accuracy drop in the middle. What is the root cause for that? The paper seems no giving any explanation. - Though the experimental results show faster convergence rate in terms of accuracy vs iteration, it's unknown whether this observation still hold true for accuracy vs time. Traditional dot product operators leverage fast matrix multiplication libraries for speedup, while there is much less support for angular operators. Minor on grammar: - Line 216, "this problem will automatically solved" -> "be solved" - Line 319, "The task can also viewed as a ..." -> "be viewed"

Reviewer 3



In this paper, a new hyperspherical convolution framework is proposed by using the angular representations. It is different from CNNs and uses SphereConv as its basic convolution operator. Experiments are conducted to show the effectiveness of the proposed method. However, there are some concerns to be addressed. First, about the Signoid SphereConv, how to choose the value of k? Is there any suggestion? Second, how to choose the SphereConv operator. To achieve better results, do we have to try each SphereConv operator? Compared to the original conv, how about the complexity of the proposed conv? From the results, the improvement of the proposed conv is somewhat limited.